# Reduced Arbuscular Mycorrhizal Fungi (AMF) Diversity in Light and Moderate Fire Sites in Taiga Forests, Northeast China

**DOI:** 10.3390/microorganisms11071836

**Published:** 2023-07-19

**Authors:** Zhichao Cheng, Song Wu, Jun Du, Yongzhi Liu, Xin Sui, Libin Yang

**Affiliations:** 1Key Laboratory of Biodiversity, Institute of Natural Resources and Ecology, Heilongjiang Academy of Sciences, Harbin 150040, China; chengzc928@163.com; 2Science and Technology Innovation Center, Institute of Scientific and Technical Information of Heilongjiang Province, Harbin 150028, China; wusong0927@126.com; 3Heilongjiang Huzhong National Nature Reserve, Huzhong 165038, China; jxnl0123@163.com (J.D.); zrbhj_2000@163.com (Y.L.); 4Engineering Research Center of Agricultural Microbiology Technology, Ministry of Education & Heilongjiang Provincial Key Laboratory of Ecological Restoration and Resource Utilization for Cold Region & Key Laboratory of Microbiology, College of Heilongjiang Province, School of Life Sciences, Heilongjiang University, Harbin 150080, China

**Keywords:** arbuscular mycorrhizal fungi, diversity, Greater Khingan Mountains, *Larix gmelinii* forest, fire sites

## Abstract

Forest fires are an important disturbance factor in forest ecosystems, and obviously change the soil environment. Arbuscular mycorrhizal fungi, as a medium and bridge between vegetation and soil, play a crucial role in mediating plant nutrient uptake and regulating the productivity, stability, and succession of vegetation–soil systems. To investigate the effects of forest fires on the community structure and diversity of arbuscular mycorrhizal fungi in cold-temperate *Larix gmelinii* forests, we collected soils from light, moderate, and heavy fire disturbance forests and a natural forest as a control forest in Greater Khingan *Larix gmelinii* forests, in the northeast of China. The community structure and diversity of arbuscular mycorrhizal fungi was sequenced using Illumina MiSeq technology and we analyzed the correlation with the soil physicochemical characteristics. The results showed that the contents of microbial biomass content (MBC), moisture content (MC), total nitrogen (TN), and available phosphors (AP) increased significantly (*p* < 0.05) with increasing fire intensity (from Light to heavy fire), but available potassium (AK) decreased significantly (*p* < 0.05). These changes were not significant. A total of 14,554 valid sequences from all sequences were classified into 66 ASVs that belonged into one phylum, one order, four families, and four genera. The genera included *Glomus*, *Ambispora*, *Paraglomus*, and *Acaulospora*, and *Glomus* was the dominant genus (the genera with the five most relative abundances) in the control and heavy-fire forests. Non-metric multidimensional scaling (NMDS) analysis showed that forest fires significantly affected the community structure of arbuscular mycorrhizal fungi (*p* < 0.01). Redundancy analysis (RDA) showed that MBC, SOC, and AP contents significantly affected the composition structure and diversity of arbuscular mycorrhizal fungi communities. This study indicated that forest fires affected the composition and diversity of soil arbuscular mycorrhizal fungi communities through changing the soil physicochemical parameters (MBC, SOC, and AP) in cold-temperate *Larix gmelinii* forests. The study of soil physicochemical properties and arbuscular mycorrhizal fungi diversity in cold-temperate *Larix gmelinii* forests in the Greater Khingan Mountains after forest fires provides a reference basis for the revegetation and reconstruction of fire sites.

## 1. Introduction

Forest fires are one of the most important environmental factors affecting biodiversity and play a crucial role in maintaining the structure and function of forest ecosystems [1]. The impact of forest fires on soils is due to the high temperature, residual ash, and alteration of the original soil matrix and microclimate conditions, resulting in drastic changes in the soil physicochemical factors and causing significant disturbances in the soil environment [2]. Forest fires are capable of releasing elements from the soil in gaseous form and erosion [1], causing changes in the rate and flow of nutrient cycling in the soil [3]; forest fires alter the soil and also affect the composition and community dynamics of microorganisms [4], with consequent changes in the rate of decomposition and return of litter [5] and the function of forest ecosystems [6].

Arbuscular mycorrhizal fungi are abundantly distributed in the soil [7] and can form reciprocal symbioses with most plant roots in the ecosystem [8]. On the one hand, they obtain nutrients from plants to complete their own growth and development [9]; on the other hand, they can also improve the host plant, improve the root surface area and inter-root absorption range of the host plant, and increase the ability to absorb water and nutrients from the soil, playing an important role in plant growth and development [5]. Previous studies showed that environmental factors obviously affected arbuscular mycorrhizal fungi growth and distribution [10,11,12]. Forest fires, as an important disturbance factor in forest ecosystems, have a profound effect on arbuscular mycorrhizal fungi composition and function though changing soil nutrients, e.g., forest fires have a significant effect on arbuscular mycorrhizal fungi infestation rate, species composition, density, and reproduction [13,14,15,16]. Moura [15] found that arbuscular mycorrhizal fungi composition and diversity was influenced by soil pH and other nutrients; Pattinson [17] found that forest fires reduced arbuscular mycorrhizal fungi density in the top soil layer. After forest fires, arbuscular mycorrhizal fungi mycorrhizal symbionts can change the root morphology and improve the nutritional status of plants, thus promoting the growth and development of host plants, improving stress and disease resistance [18,19], participating in many physiological metabolic processes of plants [20], and indirectly contributing to forest restoration by regulating the inter-root microenvironment and microbial community structure of plants [21,22].

Forest fires can alter the structure and diversity of arbuscular mycorrhizal fungi communities, but the response of arbuscular mycorrhizal fungi communities was not consistent in different geographical and climatic conditions, as well as forest types and soil conditions. Lots of studies on forest fires affecting soil arbuscular mycorrhizal fungi have been carried out in temperate and tropical forests, such as in Brazil [15], Australia [17], and Iran [16], but studies on soil arbuscular mycorrhizal fungi in different forest fire intensities in cold-temperate *Larix gmelinii* forests are still lacking. Therefore, we set forests with different forest fire intensities in Huzhong National Nature Reserve of China to investigate the changes in soil physicochemical properties and arbuscular mycorrhizal fungi community structure diversity under different fire intensities.

Huzhong National Nature Reserve is located in the Greater Khingan Mountains of China, and is a typical representative of the cold-temperate forest ecosystem, where *Larix gmelinii* forests are the climax zonal vegetation. Forest fires caused by lightning strikes are frequent in this reserve and cause forest succession and soil environment changes. At present, studies on post-fire forestry in the Greater Khingan Mountains have mainly focused on forest carbon storage [23], the restoration of above-ground vegetation [24], effects of soil bacterial communities [25], and soil black-carbon content [26]. However, studies on the changes of the structural composition of arbuscular mycorrhizal fungi communities and soil physicochemical properties on arbuscular mycorrhizal fungi diversity in fire sites are still limited. Therefore, we used high-throughput technology to sequence soil arbuscular mycorrhizal fungi in Huzhong National Nature Reserve, and analyzed the changes in arbuscular mycorrhizal fungi community structure and diversity with different fire intensities. The aim was to investigate the influence of forest fires on the structure and diversity of arbuscular mycorrhizal fungi communities in conifer forest soils, to disclose the key factors affecting arbuscular mycorrhizal fungi in different fire intensities, and to provide a reference basis for forest revegetation and reconstruction during fire disturbance.

## 2. Materials and Methods

### 2.1. Sample Site Overview

This study was located at Huzhong National Nature Reserve (122°12′16.3″–122°21′7.8″ E, 53°26′30.6″–53°28′6.3″ N; Figure 1) in the Greater Khingan Mountains, which is the largest cold-temperate *Larix gmelinii* forest ecosystem in China. It is a continental monsoon climate with an average temperature of −2 to −5.5 °C and a record low of −52.3 °C [27]. The average annual precipitation is 480 mm, and the rainfall period is mostly concentrated in July–August, accounting for about 50–60% of the annual precipitation, with a frost-free period of about 130 d and an average altitude of 600–800 m [28]. The sample sites were selected from flat terrain undisturbed by anthropogenic activities and with complete vegetation restoration. The sample sites were selected from the fire sites with flat topography, undisturbed by human activities and with intact vegetation recovery.

### 2.2. Sample Plot

Light- (L), Moderate- (M), and Heavy (H)-fire forests were selected in the Huzhong Reserve in 2010, and adjacent forests with the same stand type without disturbances from fire were selected as control forests. Each forest set up three replicates and an area of 20 m × 20 m for each forest. The basic information of each forests is shown in Table 1.

### 2.3. Soil Sample Collection and Experimental Method

Ten soil samples were collected from 0 to 10 cm in each forest after removing humus and then mixed together into one soil sample (approximate 500 g per plot). The mixed soil samples passed through 2 mm sieve to remove plant roots and stones, and then placed in ziplock bags, backstroked in an ice box, and immediately brought to the laboratory. The soil samples were divided into two sub-parts in the laboratory, one stored at −80 °C for DNA extraction and another for determination of soil physicochemical properties.

### 2.4. Soil Physical and Chemical Properties Analysis

Soil microbial biomass carbon (MBC) was determined using the carbon and nitrogen analyzer method [30], soil pH was determined using the potentiometric method [31], soil moisture content (MC) was determined with the drying method [32], soil available potassium (AK) was determined with the ammonium acetate leaching flame photometer method [33], Elementarvario ELIII (Germany) automatic carbon and nitrogen analyzer was used to determine soil organic carbon (SOC) and total nitrogen (TN) contents [34], soil available phosphorus (AP) was determined using the molybdenum antimony anti-colorimetric method [33], and soil alkaline available nitrogen (AN) was determined with the alkaline diffusion method [33].

### 2.5. DNA Extraction and Arbuscular Mycorrhizal Fungi Sequencing

Total DNA was extracted from fresh soil samples by weighing 0.25 g and following the procedure of E.Z.N.A.^®^ Soil DNA Kit DNA (Omega Bio-Tek, Norcross, GA, USA). Using diluted genomic DNA as a template, specific primers with Barcode AML1F (5′-ATCAACTTTCGATGGTAGGATAGA-3′) and AML2R (5′-GAACCCAAACACTTTGGGTTTCC-3′) were selected as the first pair of primers; AMV4.5NF (5′-AAGCTCGTAGTTGAATTTCG-3′) and AMDGR (5′-CCCAACTATCCCTATTAATCAT-3′) were used as the second pair of primers to amplify the fungal 18S rRNA region. In the second round of PCR, the barcode primers (AMV4.5NF/AMDGR) were used to distinguish the different PCRs. The PCR reaction volumes were: 10 × Buffer 2.5 μL, template DNA (20 ng·L^−1^) 2.0 μL, dNTPs (2.5 mmol·L^−1^) 1.5 μL, specific dNTPs (2.5 mmol·L^−1^) 1.5 μL, Tag enzyme 1.0 μL. The PCR reaction conditions were as follows: 95 °C for 3 min; 95 °C for 30 s, 55 °C for 30 s, 72 °C for 45 s, 30 cycles; 72 °C for 10 min. The amplified PCR products were quantified using Picogreen fluorometer, homogenized and mixed, and then sequenced by IlluminaMiseq on PE300 platform with 3 replicates for each sample. The raw data were uploaded to NCBI SRA database (sequence number: PRJNA945502).

### 2.6. Data Processing

The raw sequences were identified using Qiime2 [35] software, the raw downstream data of sequencing were initially screened according to sequence quality, the double-end sequences that passed the initial quality screening were pairwise concatenated according to overlapping bases using the software FLASH (V1.2.7), and the DADA2 [36] plug-in was used to optimize the sequences after quality control splicing. These were subjected to noise reduction, and species annotation was performed using the Naive bayes classifier with the Maarj AM database to obtain taxonomic information and to identify arbuscular mycorrhizal fungi species. The diversity cloud analysis platform (Qiime2 process) (cloud.majorbio.com, accessed on 13 March 2023) of Shanghai Meiji Biomedical Technology Co., Ltd. (Shanghai, China) was used for subsequent data analysis.

Microbial community alpha diversity was characterized using Chao1 index (Chao1 richness estimator), Shannon–Wiener index (Shannon–Wiener index), and Simpson index (Simpson index) [34]. Data were organized and analyzed using Excel 2003 and SPSS 17.0. Two-sample *t*-test was used to analyze alpha diversity, NMDS ranking analysis based on Bray–Curtis distance was used to analyze beta diversity, and RDA analysis and Pearson’s method were used for environmental factor association analysis. All statistics were conducted with the “microeco” package R language and the figures were also finished using the “ggplot2” R language.

## 3. Results

### 3.1. Analysis of the Difference in Physicochemical Properties of Soils in the Fire Intensities

The soil physicochemical properties (MBC, SOC, pH, AK, TN, SOC, AP, and AN) changed significantly in different fire intensities (*p* < 0.05) compared with the control forest (Table 2). Soil MBC, SOC, and MC in Light, moderate, and heavy fire intensities were higher than those in control forest; soil AK in Light, moderate, and heavy fire intensities was significantly lower compared with that in the control forest. Soil TN in Light and moderate did not change significantly compared to the control forest and only that in heavy fire intensities significantly increased compared with that in the control forest (*p* < 0.05).

### 3.2. Differential Analysis of Soil Arbuscular Mycorrhizal Fungi Diversity in the Fire Sites

As shown in Figure 2, Light, moderate, and heavy fire intensities significantly decreased the Chao1 and Shannon indices of soil arbuscular mycorrhizal fungi compared with those in the control forest (*p* < 0.05), but increased the Simpson index (*p* < 0.05) in the three fire intensities. Moreover, the Chao1, Shannon, and Simpson indices of soil arbuscular mycorrhizal fungi did not change within the different fire intensities (*p* > 0.05) (Figure 2).

The result of NMDS analysis is shown in Figure 3, and the stress value was 0.042, lower than 0.05, indicating that the result was well representative. The Adonis analysis showed that the arbuscular mycorrhizal fungi community structures were significant different (R = 0.9475, *p* < 0.01) in different forest fire intensities. From the Figure 3, heavy fire intensity was relatively distant from Light and moderate, where the overlap between the elliptical areas of L (Light-fire) and M (Moderate-fire) forests was higher, indicating that the diversity and composition of arbuscular mycorrhizal fungi were more similar under these two treatments, and also indicating that fire of different intensities had significant effects on arbuscular mycorrhizal fungal community composition.

As shown in Figure 4, the Bray–Curtis distance showed that the minimum distance of L (Light fire) and M (Moderate fire) was 0.56, indicating that the arbuscular mycorrhizal fungi community composition of L (Light fire) and M (Moderate fire) was similar, but the distance between CK (Control—blank) and L (Light fire), M (Moderate fire), and H (Heavy fire) groups was between 0.97 and 0.99, indicating that the arbuscular mycorrhizal fungi community composition between these fire treatments was significantly different.

### 3.3. Analysis of Differences in the Structural Composition of Arbuscular Mycorrhizal Fungi Communities in Fire Trails

As seen in Figure 5, the total number of ASVs of the different fire intensities were 20 (Light fire), 16 (Moderate fire), and 20 (Heavy fire), which was lower than that in the control forest, and the numbers of ASVs specific to the forest were L (Light fire) > M (Moderate fire) > H (Heavy fire). The number of ASVs jointly owned with the control forest between fire groups were M (Moderate fire) > H (Heavy fire) > L (Light fire), the highest number of ASVs were jointly owned by L (Light fire) and M (Moderate fire) between fire groups, and the lowest number of ASVs were jointly owned by M (Moderate fire) and H (Heavy fire).

From Figure 6 and Appendix A, the dominant genus of arbuscular mycorrhizal fungi in the fire sites of different intensities was *Glomus* (91.6~94.76%), followed by *Ambispora* (13.37~13.86%). The relative abundance of *Glomus* decreased in the fire group compared with the control group, while the relative abundance of *Glomus* increased after decreases between fire groups. In addition, *Glomus* was not detected in M (Moderate fire), and *Acaulospora* was detected only in CK (Control—blank). This shows that the community structure of L (Light fire) and M (Moderate fire) was more affected after the forest fire.

### 3.4. Correlation Analysis of Factors Influencing the Structure and Diversity of Arbuscular Mycorrhizal Fungi Communities at the Fire Sites

The results of Pearson correlation analysis (Table 3) showed that the Chao1 diversity index was significantly negatively correlated with AP (*p* < 0.05), the Shannon diversity index was significantly negatively correlated with MBC (*p* < 0.01), and the Simpson diversity index was significantly positively correlated with MBC (*p* < 0.01) and negatively correlated with AK (*p* < 0.01), indicating that AP, MBC, and AK are important soil environmental factors affecting arbuscular mycorrhizal fungi alpha diversity.

Redundancy analysis (RDA) was performed on the soil physicochemical properties for the community composition of arbuscular mycorrhizal fungi at the ASV. The first and second axes explained 91.91% and 2.16%, respectively, with a total explanation of 94.07% (Figure 7). The arbuscular mycorrhizal fungi communities of L (Light fire) were positively correlated with AP, SOC, pH, MC, and MBC and negatively correlated with TN, AK, and AN. Arbuscular mycorrhizal fungi communities of M (Moderate fire) were positively correlated with AP, SOC, pH, and MBC, and negatively correlated with MC, TN, AK, and AN. Arbuscular mycorrhizal fungi communities of CK (Control—blank) and H (Heavy fire) were positively correlated with MC, TN, AK, and AN and negatively correlated with AP, SOC, pH, and MBC.

As shown in Table 4, soil AP, SOC, and MBC had significant (*p* < 0.05) effects on soil arbuscular mycorrhizal fungi community composition, indicating that AP, SOC, and MBC were important factors affecting soil arbuscular mycorrhizal fungi communities.

Heatmap analysis of the correlation between genus and environmental factors at the genus level showed that *Ambispora* was significantly negatively correlated with soil MC content (*p* < 0.05) and significantly negatively correlated with soil TN content and soil AP content (*p* < 0.01), indicating that MC, TN, and AP were important factors affecting the genus composition of soil arbuscular mycorrhizal fungi(Figure 8).

## 4. Discussion

### 4.1. Effect of Fire on Soil Nutrient Content

After forest fires, some physical and chemical properties of forest soils, such as pH and organic carbon, were affected [37]. This result showed that there were significant differences in soil physicochemical properties between different fire intensities (Table 1), and the trend of increasing and then decreasing MBC and SOC with increasing fire intensity was consistent with previous reports [5,38], which may be due to the fact that moderate and Light fires can increase soil temperature, promote soil microbial activity, and accelerate soil organic matter formation [39], while high temperatures under heavy fires cause thermal decomposition of surface soil SOC, resulting in a decrease in organic matter input and, thus, a decrease in SOC content. As the intensity of a forest fire increases, the fire changes from a disturbance to a destructive factor and plant roots undergo apoptosis, leading to the disappearance of mycorrhizae, which reduces soil MBC and makes SOC depleted in gaseous form [40].

After the Light-intensity forest fires, the surface was exposed due to the burning of the surface litter and vegetation, which led to enhanced external erosion of the soil, causing weathering and scouring under the action of rainwater, resulting in a significant decrease in soil AK content [41]. The results of this study are consistent with the results of previous related studies [42]. In contrast, there was no significant difference between moderate and heavy fires in the results of this study, which may be due to the loss of K through particulate or non-particulate matter during combustion, and there was no longer enough AK released [43].

In the present study, soil AP content was elevated in all the fire sites, which is consistent with the results of a previous [44] related study, due to the positive response of soil from the conversion of some organic phosphorus compounds into inorganic phosphorus caused by forest fires [45,46].

The significant increase in soil MC changes, contrary to the results of Hart [1,47], could be attributed to the massive plant mortality due to the collapse of trees after forest fires. On the one hand, it can reduce plant transpiration and rainwater retention by the canopy [48], and on the other hand, with the decomposition of fallen trees, the density of fallen trees becomes smaller and the porosity increases to absorb more water [49], so it causes the soil MC value to increase.

### 4.2. Effect of Fire on Soil Arbuscular Mycorrhizal Fungi Diversity

It has been shown that fire has a large effect on the alpha diversity of arbuscular mycorrhizal fungi [50]. In the present study, we found that the arbuscular mycorrhizal fungi Chao1 index of forest soil gradually increased with the increase in fire intensity (Figure 2). There may be four reasons: Firstly, arbuscular mycorrhizal fungi have a faster growth rate [50,51]. Although fire reduces its biomass, the faster growth rate can make arbuscular mycorrhizal fungi recover quickly or even exceed the previous biomass; Secondly, forest fires can accelerate the decomposition of apoplastic matter, releasing more nutrients; Thirdly, forest fires can accelerate the decomposition of apoplankton, releasing more nutrients to arbuscular mycorrhizal fungi, resulting in a rapid increase in arbuscular mycorrhizal fungi biomass [52,53]; Fourthly, forest fires change soil physicochemical properties to indirectly affect arbuscular mycorrhizal fungi survival [16], and it has been suggested that the alpha diversity of arbuscular mycorrhizal fungi is significantly correlated with soil physicochemical properties, which is consistent with the results of this study [53,54]. In this study, Shannon and Simpson indices were found to be significantly correlated with MBC and AK, suggesting that changes in alpha diversity may be influenced by soil physicochemical factors. The Simpson index of arbuscular mycorrhizal fungi was the highest in moderate fire, while there was no significant difference between the Simpson index of heavy and Light fire (*p* > 0.05); this might be because the MBC values of the two soils of heavy and Light fire were closer, providing a more similar soil nutrient environment for the development and reproduction of arbuscular mycorrhizal fungi, resulting in no significant difference in the diversity of arbuscular mycorrhizal fungi.

Many studies have shown [16,55] that arbuscular mycorrhizal fungi abundance and diversity vary with soil nutrient availability, and in the present study, we found that AK, MBC, and AP directly influenced the alpha diversity of arbuscular mycorrhizal fungi in soils with different fire intensities and were the main drivers, which is consistent with previous studies [16,54,55]. In our study, we found no significant correlation between the Chao1 index and Shannon index of arbuscular mycorrhizal fungi in soils with different intensities of fire (Figure 2, *p* > 0.05), which may be due to the wide ecological niche of clumping mycorrhizal fungi, which can rapidly adapt to environmental changes under different fire levels [56,57].

### 4.3. Effect of Different Intensities of Fire on the Structural Composition of Arbuscular Mycorrhizal Fungi Communities

It has been shown that fire can change the fungal species composition and promote the dominance of specific species or enhance the spectrum [15]. In the present study, *Paraglomus* were all present in soils burned at different intensities of fire, while *Glomus*, *Ambispora*, *Acaulospora*, and *Archaeospora* were not evenly distributed (Figure 6), which is consistent with other studies [58]. The most abundant *Glomus* was due to its strong environmental adaptability and unique reproductive characteristics, with a high spore production rate that allowed it to colonize different environments and be highly adaptable [59]. Their lighter relative abundance in the soil of moderate to Light fire sites may be due to the fact that Light fires reduce the number of *Glomus* propagules surviving on the soil surface [17], decreasing the species abundance and density of *Glomus* [16]; the heat increased during Moderate fire and was transferred to the soil, leading to the death of *Glomus* by scorching [60], while in the case of Heavy fire, a large amount of soil surface humus was burned out, which released a large amount of nutrients and was able to provide *Glomus* with the nutrients needed for growth and accelerate its growth [61].

It has been shown that physicochemical properties such as SOC, C/N, and MBC have significant effects on fungal community composition [62,63,64], and similar results were observed in the present study, where the community structure composition of arbuscular mycorrhizal fungi in the soil of the fire sites was significantly or highly correlated with MBC, SOC, and AP. The relative abundance of *Ambispora* was negatively correlated with MC, TN, and AP, probably because *Ambispora* is more sensitive to changes in soil physicochemical properties [65], which is consistent with the results reported in the literature [27,65,66].

## 5. Conclusions

Different intensities of fire affected both the community structure and diversity of soil arbuscular mycorrhizal fungi. Fire significantly reduced the alpha diversity of arbuscular mycorrhizal fungi in cold-temperate forest soils, and MBC, SOC, and AP were the main influencing factors in soil physicochemical properties. Fire altered the beta diversity of arbuscular mycorrhizal fungi in cold-temperate forest soils, *Glomus* was the dominant taxon in the arbuscular mycorrhizal fungi of the fire sites, and MC, TN, and AP were important soil physicochemical factors affecting the community structure composition of arbuscular mycorrhizal fungi.

## Figures and Tables

**Figure 1 microorganisms-11-01836-f001:**
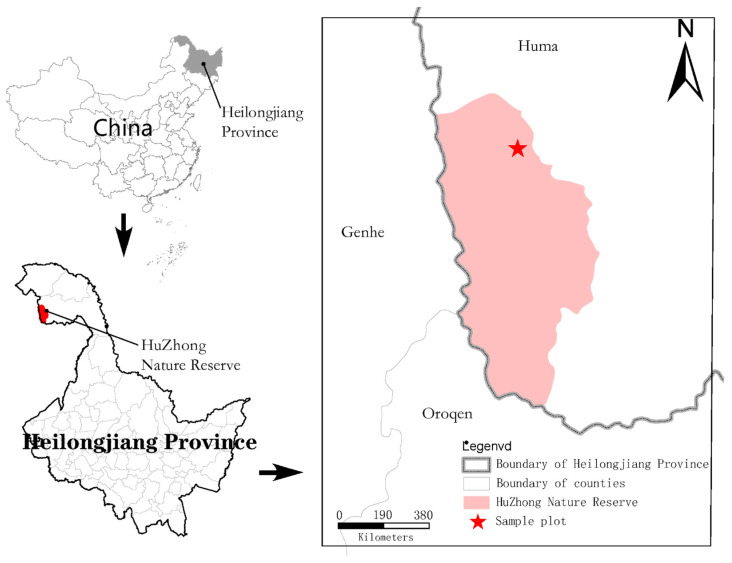
The asterisk indicates the study site in Heilongjiang Province and China.

**Figure 2 microorganisms-11-01836-f002:**
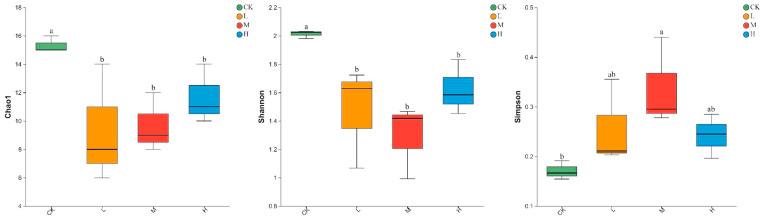
Alpha diversity of arbuscular mycorrhizal fungi communities in soils with different fire intensities. All results are reported as mean ± standard deviation (n = 3). Different letters within a row indicate significant differences (*p* < 0.05; ANOVA) among the different intensities of fire in this study.

**Figure 3 microorganisms-11-01836-f003:**
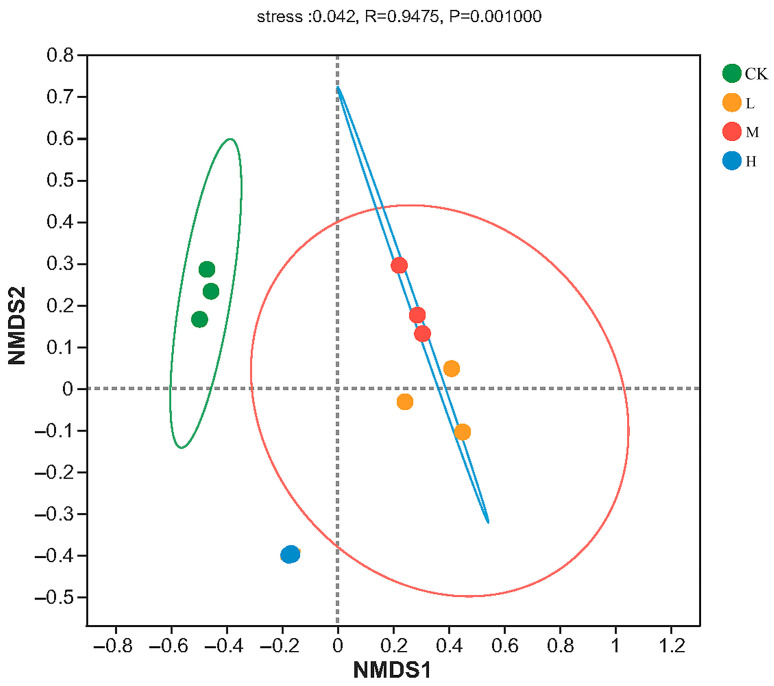
NMDS plot of the soil arbuscular mycorrhizal fungi ASV based on Bray–Curtis metrics among all samples. CK, Control—blank; L, Light fire; M, Moderate fire; H, Heavy fire.

**Figure 4 microorganisms-11-01836-f004:**
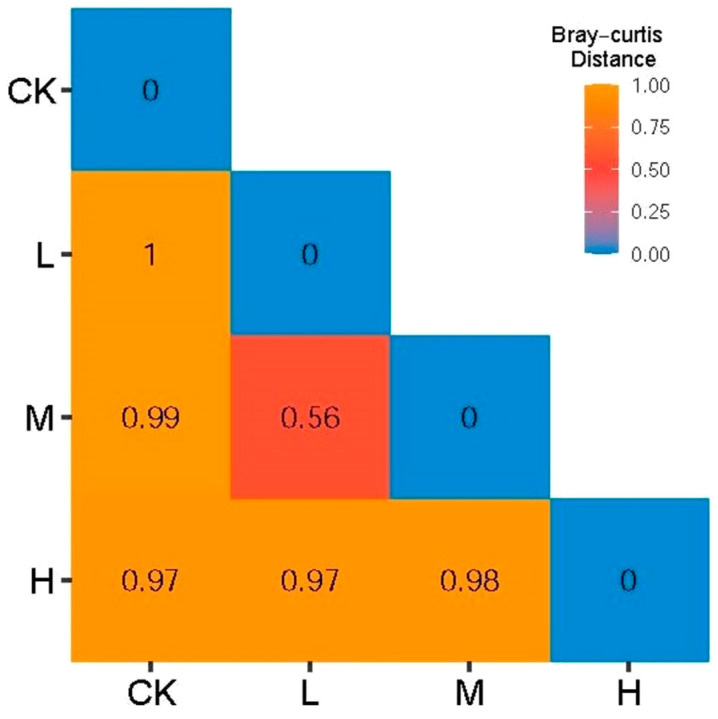
Bray–Curtis distance between soil arbuscular mycorrhizal fungi communities.

**Figure 5 microorganisms-11-01836-f005:**
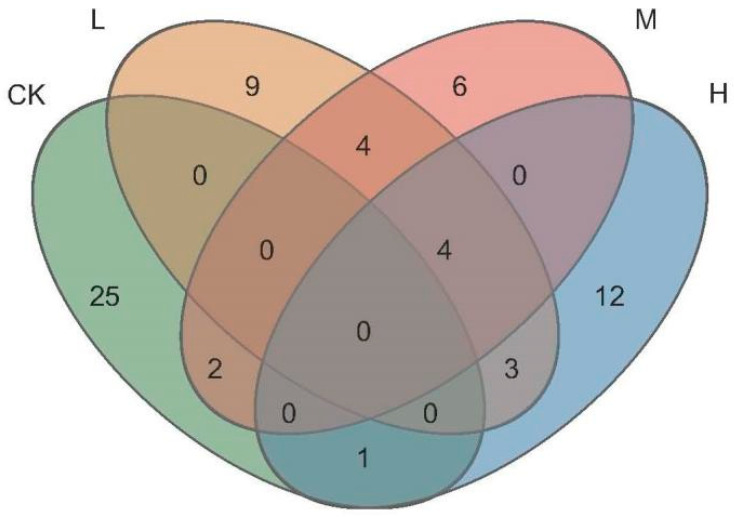
Venn diagram of arbuscular mycorrhizal fungi ASVs. CK, Control—blank; L, Light fire; M, Moderate fire; H, Heavy fire.

**Figure 6 microorganisms-11-01836-f006:**
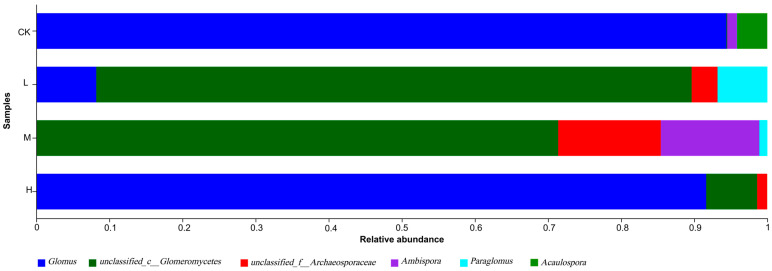
Genus-level composition of arbuscular mycorrhizal fungi detected in the forest soil at different latitudes. CK, Control—blank; L, Light fire; M, Moderate fire; H, Heavy fire.

**Figure 7 microorganisms-11-01836-f007:**
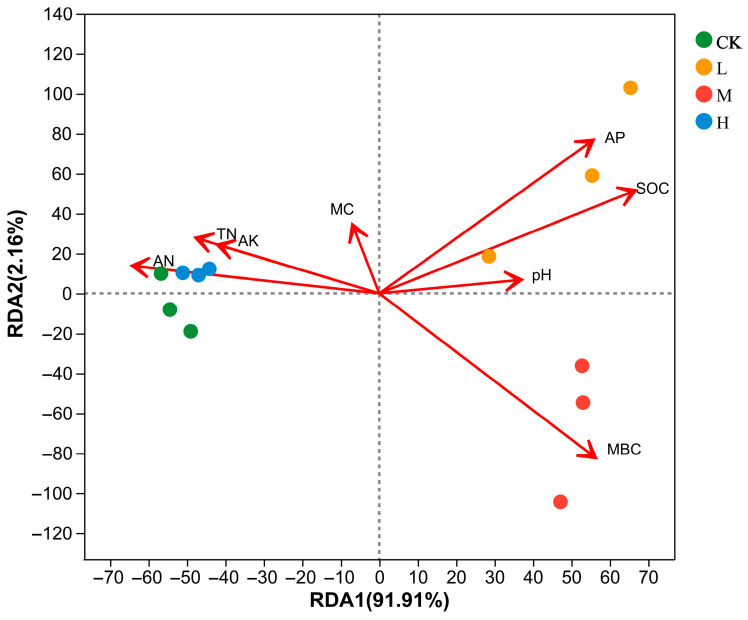
Correlations between soil physicochemical properties and arbuscular mycorrhizal fungi Genus (RDA). CK, Control—blank; L, Light fire; M, Moderate fire; H, Heavy fire.

**Figure 8 microorganisms-11-01836-f008:**
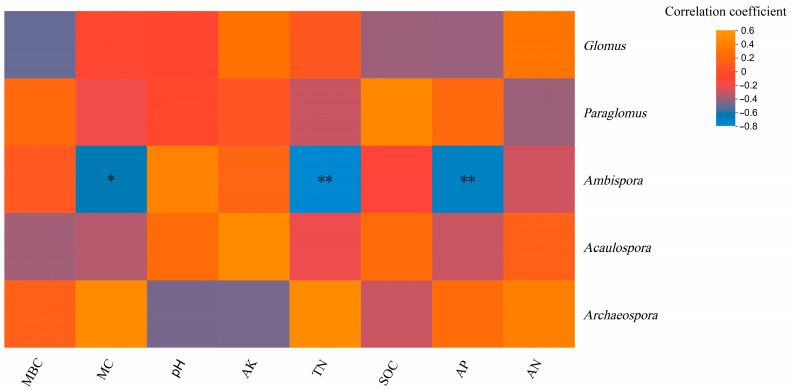
Correlation heatmap of arbuscular mycorrhizal fungi Genus and soil physicochemical properties. Note: the *x*-axis and *y*-axis are soil environmental and species, respectively. The *p*-value is shown in different colors in the figure. If the *p*-value is less than 0.05, it is marked with an * sign, * 0.01 < *p* ≤ 0.05, ** 0.001 < *p* ≤ 0.01.

**Table 1 microorganisms-11-01836-t001:** Basic situation of fire sites with different intensities. L, M, and H indicate Light, moderate, and heavy fire intensities, respectively [29].

Fire Intensity	Burning Dead Wood Ratio	Changes in Surface Vegetation
L (Light fire)	≤30%	Only some of the vegetation was burned
M (Moderate fire)	30–70%	Trees were partially burned, shrubs, herbs, and other vegetation were completely burned
H (Heavy fire)	≥70%	Trees, shrubs, herbs, and other vegetation were completely burned to death

**Table 2 microorganisms-11-01836-t002:** Physical and chemical properties of arbuscular mycorrhizal fungi in forest soils with three different fire intensities. All results are reported as mean ± standard deviation (n = 3). Different letters within a row indicate significant differences (*p* < 0.05; ANOVA) among the different intensities of fire in this study.

Intensity	MBC mg/kg	MC%	pH	AK mg/kg	TN g/kg	SOC g/kg	AP mg/kg	AN mg/kg
CK (Control—blank)	149.97 ± 33.56 c	0.04 ± 0.004 c	6.79 ± 0.03 a	213.05 ± 19.19 a	1.69 ± 0.01 b	97.05 ± 3.41 b	11.74 ± 0.69 c	96.14 ± 11.33 b
L (Light fire)	268.37 ± 23.67 b	0.16 ± 0.002 b	6.64 ± 0.02 a	140.07 ± 5.2 b	1.95 ± 0.06 b	122.98 ± 5.96 a	46.9 ± 0.83 a	89.37 ± 12.31 b
M (Moderate fire)	847.07 ± 369.5 a	0.12 ± 0.006 b	6.53 ± 0.05 a	80.43 ± 6.38 c	1.71 ± 0.06 b	93.16 ± 6.49 b	18.21 ± 0.66 c	82.13 ± 24.22 b
H (Heavy fire)	349.99 ± 12.5 b	0.27 ± 0.03 a	5.84 ± 0.4 b	80.97 ± 11.17 c	2.96 ± 0.43 a	76.06 ± 3.24 c	27.48 ± 1.57 b	148.16 ± 17.29 a

**Table 3 microorganisms-11-01836-t003:** Correlation coefficients of arbuscular mycorrhizal fungi alpha diversity indices and soil chemical factors. If the *p*-value is less than 0.05, it is marked with an * sign, * 0.01 < *p* ≤ 0.05, ** 0.001 < *p* ≤ 0.01.

Name	MBC	MC	pH	AK	TN	SOC	AP	AN
Chao1	−0.531	−0.422	0.16	0.446	−0.4078	−0.109	−0.629 *	−0.134
Shannon	−0.741 **	−0.348	0.228	0.692 *	−0.321	0.021	−0.416	0.006
Simpson	0.727 **	0.369	−0.179	−0.734 **	0.251	−0.007	0.391	−0.119

**Table 4 microorganisms-11-01836-t004:** Significance tests between soil physicochemical properties and arbuscular mycorrhizal fungi community structures. If the *p*-value is less than 0.05, it is marked with an * sign, * 0.01 < *p* ≤ 0.05, ** 0.001 < *p* ≤ 0.01.

Soil Factors	R^2^	*p*-Value
MBC	0.7839	0.004 **
MC	0.0871	0.681
pH	0.1062	0.604
AK	0.1806	0.414
TN	0.2325	0.3
SOC	0.5492	0.035 *
AP	0.7023	0.006 **
AN	0.3329	0.158

## Data Availability

Data are contained within the article.

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
