# Peer review of "Reduced Arbuscular Mycorrhizal Fungi (AMF) Diversity in Light and Moderate Fire Sites in Taiga Forests, Northeast China"

_microorganisms, 2023, doi:10.3390/microorganisms11071836_

Round 1

Reviewer 1 Report

The manuscript by Cheng et al describes the diversity of arbuscular mycorrhizal fungi in soil of sites that underwent different degrees of burning. The study is well done but the manuscript should be improved before publication:

-        The material and methods part on AMF sequencing should be expanded. What was targeted in the PCR reaction? Where whole genomes sequenced or were specific sequences targeted? Which primers were used?

-        The paper uses an unusually high amount of abbreviations and readability could be enhanced by spelling out lesser used ones.

-        Figure legends and labels should have an increased font size.

Few minor misspellings.

Author Response

Thank you for your review. We have made the following modifications (PDF) to your feedback.

Reviewer 2 Report

forest fires have significant impacts on soil microbial populations and nutrient functioins. this article uses high-throughput sequencing technology to study the effects of different degrees of forest fires on soil AMF population and soil physicochemical properties, as well as their correlation. the results provides a reference basis for revegetation and reconstruction of fire sites. 

Author Response

Thank you for teacher's review. We have checked and revised the English grammar and spelling.

Reviewer 3 Report

Arbuscular mycorrhizal fungi are among the most common soil fungi. They form a mycorrhizal association with about 70-90% of all known vascular plants. Arbuscular mycorrhizae occur in halophilic, hydrophilic and xerophilic plants. They accompany marine, dune, forest and alpine vegetation. Among plants of economic importance, only plants of the Brassicaceae and Chenopodiaceae families contain a relatively large number of species that do not usually coexist in symbiosis with arbuscular fungi. The reviewed article is scientifically interesting. The authors presented the results of their research in an interesting way. I have no comments on the reviewed paper.

Author Response

(The authors gave the same response as above.)
